# Identification of *Glu-D1* Alleles and Novel Marker–Trait Associations for Flour Quality and Grain Yield Traits under Heat-Stress Environments in Wheat Lines Derived from Diverse Accessions of *Aegilops tauschii*

**DOI:** 10.3390/ijms231912034

**Published:** 2022-10-10

**Authors:** Ikram Elsadig Suliman Mohamed, Nasrein Mohamed Kamal, Hala Mohamed Mustafa, Modather Galal Abdeldaim Abdalla, Ashraf. M. A. Elhashimi, Yasir Serag Alnor Gorafi, Izzat Sidahmed Ali Tahir, Hisashi Tsujimoto, Hiroyuki Tanaka

**Affiliations:** 1United Graduate School of Agricultural Sciences, Tottori University, Tottori 680-8553, Japan; 2Arid Land Research Center, Tottori University, Tottori 680-0001, Japan; 3Agricultural Research Corporation, Wad Medani P.O. Box 126, Sudan; 4Faculty of Agriculture, Tottori University, Tottori 680-8553, Japan

**Keywords:** hot environment, *Aegilops tauschii*, genetic diversity, dough strength, protein content, heat-resilient genotypes, heat tolerant

## Abstract

Heat stress during grain filling is considered one of the major abiotic factors influencing wheat grain yield and quality in arid and semi-arid regions. We studied the effect of heat stress on flour quality and grain yield at moderate and continuous heat stress under natural field conditions using 147 lines of wheat multiple synthetic derivatives (MSD) containing *Aegilops tauschii* introgressions. The study aimed to identify the marker–trait associations (MTAs) for the quality traits and grain yield under heat-stress conditions and identify stress-resilient germplasm-combining traits for good flour quality and grain yield. The MSD lines showed considerable genetic variation for quality traits and grain yield under heat-stress conditions; some lines performed better than the recurrent parent, Norin 61. We identified two MSD lines that consistently maintained relative performance (RP) values above 100% for grain yield and dough strength. We found the presence of three high-molecular-weight glutenin subunits (HMW-GSs) at the *Glu-D1* locus derived from *Ae. tauschii*, which were associated with stable dough strength across the four environments used in this study. These HMW-GSs could be potentially useful in applications for future improvements of end-use quality traits targeting wheat under severe heat stress. A total of 19,155 high-quality SNP markers were used for the genome-wide association analysis and 251 MTAs were identified, most of them on the D genome, confirming the power of the MSD panel as a platform for mining and exploring the genes of *Ae. tauschii*. We identified the MTAs for dough strength under heat stress, which simultaneously control grain yield and relative performance for dough strength under heat-stress/optimum conditions. This study proved that *Ae. tauschii* is an inexhaustible resource for genetic mining, and the identified lines and pleiotropic MTAs reported in this study are considered a good resource for the development of resilient wheat cultivars that combine both good flour quality and grain yield under stress conditions using marker-assisted selection.

## 1. Introduction

Heat stress is considered one of the most significant abiotic stress factors influencing wheat flour quality and grain yield. In the face of increasing changes in global climate, understanding and diagnosing the impact of high temperature on wheat flour quality and grain yield are necessary. Moreover, assessing differential genotypic responses is crucial for identifying resilient genetic resources that combine heat tolerance and good quality.

The available literature focuses more on identifying heat-tolerant genotypes by examining the impact of heat stress on yield or yield-related traits without an in-depth analysis of the quality aspects [1,2]. Moreover, most of the studies on wheat quality have been conducted in controlled environments or with a relatively small number of genotypes [3,4,5,6]. Furthermore, the genetic basis of the diversity resilience and genome-wide association studies for wheat quality under heat stress has yet to be fully explored.

Wheat grain quality, a characteristic that affects food processing quality and nutritional value, is crucial for assessing the market potential and commercial value of new wheat varieties. One of the most important characteristics affecting wheat quality is the unique gluten protein. Gluten proteins, also called seed storage proteins (SSPs), are classified into monomeric gliadins and polymeric glutenin. The gliadin proteins are classified into four major types: α-, β-, γ-, and ω-gliadins, according to their electrophoretic mobility in acid conditions [7]. The glutenins are classified into high-molecular-weight (HMW) and low-molecular-weight (LMW) types. The gliadins are vicious, provide extensibility, and are stretchable, whereas the glutenins give elasticity to bread dough [8]. When gliadins and glutenins are mixed with water, they produce visco-elastic dough. These physical properties are associated with the functional traits of flour quality. For example, a flour dough that exhibits appropriate gas-holding properties is required for bread making, whereas a dough that exhibits weak gas-holding properties is necessary for making cookies and cakes. Therefore, the technical properties of wheat flour are directly related to the gliadin: glutenin ratio in the flour. Thus, various food products can be made depending on the specific balance of the functional properties of the dough [9]. The high-molecular-weight glutenin subunits (HMW-GSs) have a significant impact on wheat flour quality because they constitute the primary factor determining gluten elasticity, thus are important for the bread-making process [10,11]. The HMW-GSs represent about 10% of SSPs; however, almost 80% of the variations in the Alveograph baking strength (*w*) value can be attributed to variations in HMW-GS composition [12].

The genes encoding HMW-GSs are located on the long arms of chromosomes 1A, 1B, and 1D at the *Glu-A1*, *Glu-B1*, and *Glu-D1* loci, respectively [13,14]. Alleles on the *Glu-D1* locus have been reported to greatly affect the wheat flour quality [15,16]. It has been demonstrated that HMW-GS 5 + 10 at the *Glu-D1* locus has a higher positive effect on flour quality than HMW-GS 2 + 12 and other subunits. Moreover, the HMW-GSs at *Glu-D1* derived from *Aegilops tauschii* have been reported to cause wide variations in dough strength [17]. These significant effects of HMW-GSs on dough strength and other flour quality characteristics have been evaluated in crops grown under normal conditions. However, it is not well understood whether the effect of HMW-GSs on dough strength and flour quality is similar under field conditions with continuous heat stress. Heat stress has been documented to increase protein content [18,19,20,21]. A high protein content is generally used as an indicator of strong dough strength and increased bread loaf volume. However, during grain filling, high temperatures (>30–35 °C) have been reported to cause adverse impacts on bread dough strength [22] as a result of the concomitant alterations in the composition of gluten proteins [23] and the increase in the gliadin: glutenin ratio [4]. Wheat quality is controlled by many genes and quantitative trait loci (QTLs), which are significantly influenced by environmental factors [24,25]. Previous studies identified QTLs associated with GPC in almost all tetraploid and hexaploid wheat chromosomes [26,27]. However, most previous studies documented more MTAs for the protein content on the B and A genomes than on the D genome. For example, Irina et al. [28] identified eleven significant MTAs for mean protein content evaluated across six environments, of which nine were on chromosome 6A. Also, Liu et al. [29] detected QTLs for protein content on chromosomes 2B and 7B, whereas Prasad et al. [30] reported 13 QTLs on chromosomes 2A, 2B, 2D, 3D, 4A, 6B, 7A, and 7D. In the previous literature, the D genome was observed to have a relatively lower contribution to the protein content, which may be attributed to the lack of diversity in the D genome in common wheat [31].

High-temperature stress significantly restricts common wheat productivity in tropical and subtropical areas [32]. It causes morphological and physiological changes at all stages, resulting in considerable yield losses [20,33,34,35]. During flowering, high temperatures decrease grain numbers and after anthesis, temperatures above 34 °C reduce yield potential [36,37,38]. It has been documented that high temperatures cause a massive drop in grain yield of up to 46.63% [39]. A report by Asseng et al. [40] stated that an increase in temperature of 1 °C reduces grain yield by 6%. Also, temperature and yield analysis conducted in the world’s hottest wheat-growing region [41] underscore the critical need of developing climate-resilient wheat cultivars. Identifying genetic loci for grain yield is essential for yield improvement through marker-assisted selection (MAS) to develop resilient wheat cultivars. Many MTAs have been identified for grain yield on chromosomes 1A, 1B, 2A, 2D, 3A, 3B, 3D, 5A, and 5B [42], as well as on chromosomes 4B and 6B [43]. Under heat-stress conditions, MTAs on chromosomes 4A, 6A, 5B, and 3B have been identified [44,45,46,47].

Sudan is characterized as the world’s hottest wheat-growing environment [41]. Heat stress is the main abiotic stress that reduces wheat productivity in Sudan. However, yields of up to 5–6 tons/ha have been achieved thanks to tremendous research efforts in collaboration with international research centers such as CIMMYT and ICARDA. However, the average yield in farmers’ fields across the country still far exceeds (1.8–2.0 t/ha) that achieved in research for various reasons. Despite improving wheat productivity during the short season (100–110 days), the heat-stressed conditions of Sudan are a significant challenge for wheat researchers and producers [20,48]. The methodology used to meet this challenge has included the evaluation of agronomic performance and stability of promising wheat genotypes in multi-environment trials across locations with a heat-stress gradient from relatively cool northern Sudan (Dongola and Hudeiba) to hot central Sudan (Wad Medani and New Halfa). This methodology has led to the identification of many high-yielding elite lines that have adapted to the favorable irrigated areas as well as heat-stress environments [49].

Climate-resilient wheat germplasm has become rare due to the narrow genetic diversity of common wheat [50,51]. The wild relative, the D genome donor of common wheat, *Aegilops tauschii*, has been widely reported in stress-resilience breeding to expand wheat genetic diversity [50,52,53,54]. Thus, to explore the genetic diversity of *Ae. tauschii* for wheat improvement, a platform of a wheat multiple synthetic derivative (MSD) panel has been developed using 43 *Ae. tauschii* accessions [54,55]. This MSD platform successfully enabled the exploration of the wide genetic diversity of heat-stress adaptive traits [55,56] and showed high genetic variations in drought resilience-related traits [57]. Moreover, novel alleles and QTLs associated with resilience to combined heat and drought stress under natural field conditions were identified in MSD lines [58]. In addition, the kernel weight and shape-related characteristics under heat and combined heat–drought stresses were explored [59,60]. Likewise, the MSD population showed a wide range of allelic diversity at the *Glu-D1* locus and a considerable variation in dough strength due to different introgressed portions of *Ae. tauschii* [17]. Thus, this population is expected to hold genes or resilience lines for improving wheat quality under heat stress.

With a hypothesis that *Ae. tauschii* genes could enhance wheat-bread-making quality characteristics under heat-stress conditions, we conducted this study using the MSD population to (i) explore the effect of heat stress on flour quality and grain yield under moderate and continuous heat stresses in the field, (ii) identify the marker–trait associations (MTAs) significantly associated with quality and grain yield traits under heat-stress conditions, (iii) identify the stress-resilient lines that combine both grain yield and good quality traits, and (iv) assess the extent to which *Ae. tauschii* diversity can be harnessed to improve wheat quality under heat-stress conditions.

We found that the presence of certain HMW-GS alleles at the *Glu-D1* locus derived from *Ae. tauschii* was associated with relatively stable dough strength across environments ranging from optimum to severe heat-stressed conditions. We identified novel MTAs for the grain yield and flour quality traits in heat-stress environments in wheat lines derived from diverse accessions of *Aegilops tauschii*. In addition, we identified the stress-resilient lines that combined both grain yield and good quality traits.

## 2. Results

During the heading and grain-filling stages, Dongola in the 2019/2020 season (DON19/20) was the coolest, followed by Hudeiba in the 2019/2020 season (HUD19/20) and Wad Medani in the 2019/2020 season (MED19/20), whereas Wad Medani in the 2018/2019 season (MED18/19) was the hottest (Figure 1).

### 2.1. Protein Content (%)

The environmental (E) and the genotypic (G) effects for protein content were significant; however, the G × E interaction effect was not significant (Table 1). Significant differences were found among the four environments in the protein content of the MSD lines (Figure 2a). The mean protein content in the optimum environment at DON19/20 was the lowest, whereas the highest mean value was recorded at MED18/19. Compared to the coolest environment (DON19/20), the protein content increased by 11.0, 13.9, and 25.3% at HUD19/20, MED19/20, and MED18/19, respectively.

A wide range of variations was found in the protein content within each environment. In the MSD lines, the protein contents ranged from 10.19 to 16.69% at DON19/20, from 12.21 to 18.28% at HUD19/20, from 13.21 to 18.06% at MED19/20, and from 13.62 to 20.5% at MED18/19. The protein contents of the check cultivars were comparable at MED18/19, DON19/20, and HUD19/20 (Table 1). The protein contents of Imam and Goumria slightly increased under the heat-stress conditions, albeit not significantly different from the values obtained under normal conditions. For the recurrent parent of the MSD lines ‘Norin 61’ (hereafter referred to as N61), the protein content under the optimum conditions at DON19/20 was significantly higher than that under moderate heat-stress conditions at HUD19/20.

In comparison with N61, the protein contents of the 25 (17.6%), 40 (28.0%), and 69 (47.0%) MSD lines were significantly higher at DON19/20, MED19/20, and MED18/19, respectively (Appendix A). On the other hand, the protein contents of 11 (7.7%), 20 (15.0%), and 3 (2.1%) obtained from the MSD lines were significantly lower than those of N61 at DON19/20, HUD19/20, and MED19/20, respectively. The MSD lines MSD5, MSD24, MSD12, and MSD81 showed more than a 25% increase in the protein content in the moderate and continuous heat-stress environments compared to DON19/20. On the other hand, the protein content of some MSD lines (MSD160, MSD219, and MSD413) was relatively stable across the four environments.

### 2.2. Dough Strength/Specific Sedimentation Values (SSVs) (mL/%)

The separate and combined analyses revealed the highly significant effects of the genotype and G × E on dough strength. The mean dough strength at MED19/20 was significantly higher than that at DON19/20, MED18/19, and HUD19/20 (Figure 2b). The mean dough strength at HUD19/20 was significantly higher than that at DON19/20 and was comparable with that of MED18/19. The MSD lines showed highly significant differences in dough strength (*p* < 0.01) at MED18/19, MED19/20, HUD19/20, and DON19/20 (Table 1). The dough strength of the MSD lines ranged from 0.20 to 0.48 mL/% at DON19/20, from 0.18 to 0.54 mL/% at HUD19/20, from 0.13 to 0.62 mL/% at MED18/19, and from 0.20 to 0.65 mL/% at MED19/20 (Figure 2b).

The dough strength values of the three checks were comparable at MED19/20, DON19/20, and HUD19/20. Under severe heat stress at MED18/19, N61 showed a significantly lower dough strength value than that of Imam. Imam showed the highest dough strength under continuous heat-stress conditions at MED18/19 and MED19/20 among the check cultivars. Interestingly, the lowest dough strength value for Imam was found under moderate heat stress (HUD19/20), where the highest protein content value was recorded (Table 1).

Compared to N61, the dough strength of two lines (MSD65 and MSD159) was significantly higher at MED19/20. The MSD112 line maintained high dough strength in all environments (ranked among the top seven genotypes in the four environments). A number of MSD lines showed comparable dough strength values to N61 at the four locations. On the other hand, the 94, 39, 72, and 37 MSD lines showed significantly lower dough strength than N61 at MED18/19, MED19/20, DON19/20, and HUD19/20, respectively (Appendix A).

Next, we studied the effect of HMW-GSs at the *Glu-D1* locus on dough strength. Regardless of the HMW-GSs at the A and B genomes, the HMW-GS 2.2 + 12 derived from N61 had higher dough strength at MED19/20 than in other environments (Figure 3a). For the HMW-GSs derived from *Ae. tauschii*, 2^t^ + 12^t^ and 2^t^ + 10^t^ showed higher dough strength in the heat-stress environments than in the normal environment (Figure 3b,c, respectively), whereas HMW-GSs 2.1^t^ + 12^t^, 2^t^ + 12.1^t^, and 5^t^ + 10^t^ showed no significant effects in the normal and heat-stress environments (Figure 3d–f). In the hot environment (MED19/20), lines possessing these three HMW-GSs, 2.1^t^ + 12^t^, 2^t^ + 12.1^t^, and 5^t^ + 10^t^, showed slightly higher (albeit insignificant) dough strengths than in the optimum environment (DON19/20).

When we considered the three HMW-GSs in the A, B, and D genomes, the MSD lines with HMW-GS combinations of 2*, 6 + 8, 2.2 + 12; 2*, 7 + 8, 2.2 + 12; 2*, 7 + 8, 2^t^ + 12^t^; 2*, 6 + 8, 2^t^ + 10^t^; and 2*, 6 + 8, 2^t^ + 12^t^ showed higher dough strength at MED19/20 than at DON19/20. Meanwhile, the lines with HMW-GS combinations of 2*, 7 + 8, 2^t^ + 12^t^; 2*, 6 + 8, 2^t^ + 10^t^; and 2*, 6 + 8, 2^t^ + 12^t^ derived from the D genome of *Ae. tauschii* had higher dough strength under continuous heat stress at MED18/19 than under the optimum conditions at DON19/20 (Figure 4).

For the relative performance (RP), two RP values were calculated for each line: one for MED18/19 (RP1) and the other for MED19/20 (RP2). The dough strength relative performance (RP1.SSVs) at MED18/19 ranged from 51.4 to 162% with that of N61 being 97%, whereas the RP at MED19/20 (RP2.SSVs) ranged from 74.1 to 202.7% and N61 had an RP2 value of 104% (Appendix A). The MSD lines with RP.SSV values above 100% were considered highly efficient in maintaining or possessing better dough strength. Accordingly, 78 MSD lines had higher RP1.SSV values than N61. For RP2.SSVs, 118 MSD lines had higher values than N61. A total of 75 MSD lines had consistently higher RP1.SSV and RP2.SSV values than N61. Among these 75 lines, 35 MSD lines had an HMW-GS composition of 2*, 7 + 8, 2.2 + 12 of the recurrent parent N61. The remaining 40 MSD lines possessed HMW-GS compositions of 2*, 7 + 8, 2^t^ + 12^t^ (12 lines); null, 7 + 8, 2.2 + 12 (8 lines); 2*, 6 + 8, 2.2 + 12 (5 lines); 2*, 7 + 8, 2^t^ + 10^t^ (4 lines); 2*, 6 + 8, 2^t^ + 12.1^t^; 2*, 6 + 8, 2^t^ + 12^t^; and null, 6 + 8, 2.2 + 12 (2 lines for each); and null, 6 + 8, 2^t^ + 10^t^; 2*, 7 + 8, 5^t^ + 10^t^; null, 7 + 8, 5^t^ + 10^t^; and null, 6 + 8, 2^t^ + 10^t^ (one line for each). It is worth mentioning that among the 75 lines that had consistently higher RP1.SSV and RP2.SSV values, 24 lines were comparable to N61 in their dough strength at DON19/20, and the others showed significantly lower values.

### 2.3. Grain Yield (kg/ha)

The combined analysis revealed highly significant G, E, and G × E effects. Highly significant differences (*p* < 0.001) were found among the MSD lines for grain yield in all environments (Table 1). The grain yield significantly differed among the three environments (Figure 2c). The reductions in grain yield at MED19/20 and MED18/19 were 30.1 and 39.1%, respectively, compared to DON19/20. Even within the same location, high temperatures at MED18/19 caused a 12.7% reduction in grain yield compared to MED19/20.

The heat stress significantly decreased the grain yield of the two adaptive Sudanese cultivars Imam and Goumria, as well as N61. The decrease was consistent with the increases in temperature and protein contents of the cultivars. Imam showed the highest grain yield value in all conditions and its protein contents were the lowest under all conditions. Goumria showed a significant decrease even in the two stressed environments (MED19/20 and MED18/19). The grain yield of N61 was comparable to that of the Sudanese cultivar in the two heat-stress environments (MED18/19 and MED19/20), whereas it was significantly lower at DON19/20 (Table 1).

In the optimum environment at DON19/20, three MSD lines (MSD53, MSD55, and MSD222) had significantly higher grain yields than N61, whereas eight MSD lines showed lower grain yields than N61 (Appendix A). In the heat-stress environment at MED19/20, nine MSD lines had significantly higher grain yields than N61, whereas 13 MSD lines showed lower grain yields than N61.

Under the continuous heat stress at MED18/19, 10 MSD lines had significantly higher grain yields than N61. Notably, the grain yield of MSD53 at MED18/19 was significantly higher than those of even the adaptive Sudanese cultivars, Imam and Goumria. Meanwhile, three MSD lines (MSD55, MSD77, and MSD205) showed significantly lower grain yields than N61 under the heat-stress conditions at MED18/19. The highest reduction percent in grain yield at MED18/19 was recorded for MSD55 and MSD205 (77.5 and 69.4%, respectively) compared to DON19/20. At MED19/20, the highest reductions in grain yield were recorded for MSD427 (75.8%), MSD332 (73.3%), and MSD215 (70.2%) compared to DON19/20 (Appendix A).

The relative performance values of the grain yield (RP.GY) at MED18/19 relative to DON19/20 (RP1) ranged from 22.5 to 201.2% and that of N61 recorded 57.7%. On the other hand, the RP value of the grain yield at MED19/20 relative to that at DON19/20 (RP2) ranged from 24.2 to 170% and that of N61 recorded 71.0%. Seven MSD lines consistently had RP1 and RP2 values above 100% (Appendix A). Among these lines, MSD024 and MSD026 also showed RP values for dough strength above 100%. On the other hand, 14 MSD lines showed RP1 and RP2 values of less than 50%, whereas four MSD lines showed relatively stable RP values (RP1 and RP2 ranged from 90 to 110%).

Among the 75 MSD lines with an RP of dough strength consistently above 100%, six lines showed an RP for grain yield far better than that of the recurrent parent N61 (RP ranged from 90 to 130%). Five of these six MSD lines had an HMW-GS composition of 2*, 7 + 8, 2.2 + 12. The other line had an HMW-GSs combination of Null, 7 + 8, 2.2 + 12 (Appendix A).

The MSD lines showed moderate broad-sense heritability estimates for the protein content (0.68) and grain yield (0.58), whereas dough strength had a high broad-sense heritability estimate of 0.86 (Table 1).

### 2.4. Marker–Trait Associations for Protein Content

Across the four environments, we identified 43 marker–trait associations (MTAs) significantly associated with the variations in the protein content on 14 chromosomes (Appendix A).

The highest number of MTAs (33 MTAs) for protein content was detected in the optimum environment at DON19/20, which explained 10–19% of the phenotypic variation in the protein content (Figure 5a and Appendix A). It is noteworthy that 70% of them (23 MTAs) were on the D genome, of which 74% (17 MTAs) were collocated on chromosome 6D at 365.03–471.7 Mbp, which explained 10–19% of the phenotypic variation in the protein content.

Under the moderate heat-stress conditions at HUD19/20, we identified only one MTA on chromosome 3D that explained 11.8% of the phenotypic variation (Figure 5b). Under the continuous heat-stress conditions at MED18/19 and MED19/20, we found four and five significant MTAs, respectively (Figure 5c,d). At MED18/19, the MTAs explained 10–16% of the phenotypic variation, whereas, at MED19/20, they explained 9–17% (Appendix A). We did not detect any stable marker for protein content across the environments. However, at DON19/20, the MTA on chromosome 4B (at 575.3 Mbp) was close to those MTAs on the same chromosome at MED18/19 and MED19/20 (at 654.1 and 478.9–533.3 Mbp, respectively).

### 2.5. Marker–Trait Associations for Dough Strength (SSVs)

Under the optimum conditions at DON19/20, we identified five MTAs on chromosomes 1A, 1D, 2B, 2D, and 5B, explaining about 9.1–15.8% of the variation in dough strength (Figure 5e and Appendix A). Under the moderate heat-stress conditions at HUD19/20, we identified six MTAs, explaining 11.9–16.7% of the phenotypic variation (Figure 5f). Of the six MTAs identified at HUD19/20, four collocated on chromosome 6D (12.4–17.9 Mbp), and the other two were located on chromosomes 2B and 4B (Figure 5f and Appendix A). A total of 35 and 61 MTAs explaining 9.4–20% and 9.6–48.5% were identified at MED18/19 and MED19/20, respectively. At MED18/19, several MTAs were collocated on chromosomes 1A (two at 12.72–12.74 Mbp and seven at 500.2–559 Mbp), 1D (two at 412.3–421.8 Mbp and one at 6.321 Mbp), 4D (five at 4.8–26.18 Mbp and three at 86.2–123 Mbp), 6A (two at 37.5–38.4 Mbp), 6D (three at 11.02–24.03 Mbp), and 7D (three at 191.1–348.9 Mbp). At MED19/20, most of the 61 MTAs were collocated on specific chromosomes. Interestingly, some of these collocated markers’ positions overlapped with the positions of the markers detected at MED18/19, for example, some MTAs on chromosomes 1A and 1D (Figure 5g,h, and Appendix A).

We detected 18 stable MTAs at MED18/19 and MED19/20 (Table 2). Out of these 18 MTAs, 8 were on chromosome 1A (1 at 97.9 Mbp and 7 at 500–513 Mbp); 2 each were on chromosomes 1D (at 412.3–421.8 Mbp), 4D (at 4.8 and 123.0 Mbp), and 6A (at 37.5–39.4 Mbp); and 1 each on chromosomes 2A (at 697.3 Mbp), 4A (at 403.7 Mbp), 1B (at 559.0 Mbp), and 7D (at 245.7 Mbp). One of the stable markers (1055706|F|0-65) on chromosome 4D had a pleiotropic effect on grain yield in MED18/19 and SSVs.RP2. Although we did not detect stable markers under both optimum and heat-stress conditions, a region on chromosome 1D consistently possessed MTAs under both conditions. In the optimum environment at DON19/20, an MTA on chromosome 1D (at 470.8 Mbp) was close to the MTAs on the same chromosome detected at MED18/19 and MED19/20 (at 412.3–421.8 Mbp and 410.5–431.3 Mbp, respectively) (Appendix A). Similarly, the MTA on chromosome 2D at DON19/20 (at 607.9 Mbp) was close to the MTAs on the same chromosome detected at MED18/19 and MED19/20 (at 637.7 and 588.5–613.17 Mbp, respectively) (Appendix A).

### 2.6. Marker–Trait Associations for Grain Yield

A total of 53 MTAs significantly associated with grain yield were identified across 12 chromosomes (2A, 2D, 3D, 4A, 4B, 4D, 5A, 5B, 6A, 7A, 7B, and 7D) in the optimum and continuous heat-stress environments (Appendix A). We detected 27 MTAs for grain yield under the optimum conditions at DON19/20, explaining 9–20% of the phenotypic variation (Figure 5i). Out of the 27 MTAs, 52% (14 MTAs) were identified on the D genome, 30% (8 MTAs) on the A genome, and 19% (5 MTAs) on the B genome. A region containing eight MTAs on chromosome 4D located close to each other (at 22.2–29.5 Mbp) showed a strong association with grain yield and explained 9–20% of the phenotypic variation (Appendix A). Similarly, regions on chromosomes 3D and 7B showed the same trend, with three MTAs located close to each other (at 12.45–19.069 Mbp and 650.414–647.712 Mbp, respectively), showing a strong association with grain yield, and accounted for 14–17% and 10–11% of the phenotypic variations, respectively.

Under the severe heat-stress conditions at MED18/19, we identified nine MTAs, of which seven were located on chromosomes 2D (two MTAs at 13.7 and 647.8 Mbp) and 4D (one at 10.4, two at 98.4–123.08, and three at 123.01–335.2 Mbp) (Figure 5j and Appendix A). The remaining two MTAs were located very close to each other on chromosome 4B at 657.37–657.47 Mbp. The contribution of these nine MTAs to the observed phenotypic variation ranged from 9–14% (Appendix A).

At MED19/20, 17 MTAs, which explained 9–19% of the phenotypic variation, were identified (Figure 5k and Appendix A); 13 (76%) were on the D genome, 3 were on the B genome, and 1 was on the A genome. An MTA 3944774|F|0-68 on chromosome 4D had the strongest association and explained 19% of the phenotypic variance, followed by MTAs on chromosome 3D (2259412|F|0-14) and on chromosome 7D (3947097|F|0-6) that explained 17 and 16% of the phenotypic variations, respectively.

Although no stable markers for grain yield were observed across all environments, several markers were collocated on the same chromosomal regions across the different environments. The MTAs on chromosome 7B at DON19/20 (at about 647–650 Mbp) overlapped with those on the same chromosome (at 639.2–650.4 Mbp) detected at MED19/20. The MTAs detected at MED18/19 on chromosome 4D were close to those detected at MED19/20 on the same chromosome. Similarly, the MTA on chromosome 6D at 474.5 Mbp detected at DON19/20, was close to that detected at 464.8 Mbp at MED19/20 (Appendix A).

### 2.7. Marker–Trait Associations for Relative Performance (RP) of Dough Strength and Grain Yield

We conducted GWAS using the RP to identify the MTAs significantly associated with the stability of the dough strength and grain yield under heat-stress conditions. For the dough strength, we detected 35 and 5 MTAs using RP1.SSVs and RP2.SSVs, respectively. For RP1.SSVs, 10 MTAs collocated on chromosome 4D (5 at 6.8–33.8 Mbp, 3 at 99.3–152.1 Mbp, and 2 at 335.2–465.8 Mbp), explaining 13–19% of the phenotypic variation (Appendix A). Similarly, eight MTAs collocated on chromosome 5A at 466.9–654.8 Mbp and explained 14–16% of the phenotypic variation. For RP2.SSVs, out of the five MTAs, four were consistent with RP1.SSVs and were considered stable MTAs (one on chromosome 2B (at 42.9 Mbp), one on chromosome 4D (at 123.01 Mbp), and two on chromosome 6D (at 31.1 and 139.07 Mbp)) (Appendix A).

For the grain yield, we detected six MTAs using RP1 and five MTAs using RP2.GY with no consistent MTAs between both RP1 and RP2. In RP1.GY, the MTAs explained 11–17% of the phenotypic variation, whereas in RP2 they explained 12–43% of the phenotypic variation (Appendix A). The MTA detected in RP1.GY on chromosome 2A at 4.1 Mbp was close to that detected for dough strength RP1.GY at 4.2 Mbp on the same chromosome. Likewise, the MTA detected in RP2 on chromosome 3D at 14.2 Mbp was close to that detected for dough strength RP1 at 19.6 Mbp on the same chromosome.

### 2.8. Concurrent/Pleiotropic Effect

We identified nine MTAs that had a pleiotropic effect on grain yield, dough strength, the RPs of grain yield, and dough strength in different environments (Table 2). Among these nine MTAs, five were collocated on chromosome 4D, three at the distal part (at 23.4–25.7 Mbp), and two at 123.01–335.2 Mbp. These MTAs were significantly associated with grain yield, dough strength, and the RP of dough strength and could serve as potential markers in wheat molecular breeding for these traits. The MTAs on chromosome 4D (1201923|F|0-38 and 1062681|F|0-26) had a pleiotropic effect on grain yield at DON19/20 and dough strength at MED18/19, which explained about 10.47–18.27% of the phenotypic variation. An MTA on chromosome 4A (1042486|F|0-52) had a pleiotropic effect on grain yield at DON19/20 and dough strength at MED18/19, as well as on the RP1 for dough strength. Three MTAs on chromosome 4D (998809|F|0-7, 1055706|F|0-65, and 1051116|F|0-23) underlie both grain yield and dough strength in the heat -stress environment (MED18/19), as well as RP1 or RP2 for dough strength. Similarly, we identified three MTAs that control grain yield at DON19/20 and RP1 and RP2 for grain yield and RP1 for dough strength (Table 2). Although the number of the pleiotropic markers was only nine, several MTAs identified in this study collocated with other MTAs on the same chromosome regions that affect other traits.

### 2.9. Allele’s Contribution, Candidate Genes, and Gene Expression

To investigate the contributions of N61 and *Ae. tauschii* alleles to the heat-stress tolerance in each HMW-GS, the alleles of the RP1 and RP2 for dough strength were analyzed and are explained in Figure 6 and Figure 7. For RP1, the marker rs1092339 on chromosome 3D was associated with the stability/heat-stress tolerance of dough strength under heat-stress conditions in lines harboring HMW-GS 2.2 + 12 and 2^t^ + 12^t^ and SNP allele “C” from *Ae. tauschii* (Figure 6a). The results were the opposite in marker rs32025569 on chromosome 6D for the same HMW-GS (Figure 6b). The marker rs1099989 on chromosome 4D was associated with decreased dough strength due to the heat stress in lines harboring HMW-GS 2^t^ + 12^t^ and SNP allele “N” (Figure 6c). The marker rs32025569 on chromosome 6D was associated with maintaining/stability of dough strength under heat stress in lines harboring HMW-GS 5^t^ + 10^t^ and SNP allele “N” (Figure 6b); however, the same subunit (5^t^ + 10^t^) also showed good heat tolerance (RP above 80%) when it was carrying SNP allele “C” from N61 and heterozygous SNP allele “C:T”. The same trend was observed for subunit 2^t^ + 2.1^t^, where it showed a high relative performance (above 80%) when it was carrying SNP allele “T” from *Ae. tauschii* and SNP allele “C” from N61, as well as SNP allele “N”. The marker rs1100384 on chromosome 1D was associated with the heat tolerance of dough strength under heat-stress conditions in lines harboring HMW-GS 2^t^ + 12.1^t^, 5^t^ + 10^t,^ and, 2^t^ + 10^t^, and both SNP allele “G” and “A” from N61 and *Ae. tauschii*, respectively (Figure 6e).

For RP2, the marker rs1696915 on chromosome 2B was associated with maintaining dough strength under heat stress for lines harboring HMW-GS 2.2 + 12, 2.1^t^ + 12^t,^ and 2^t^ + 12.1^t^ and SNP allele “C: A” (Figure 7a). However, the subunit 2^t^ + 12.1^t^ also maintained dough strength when it carried the SNP allele “C” from N61 and SNP allele “N”. The marker rs986590 on chromosome 6D was associated with stabilizing dough strength under heat stress for lines harboring HMW-GS 2.2 + 12 and 2^t^ + 12^t^ and SNP alleles “N” (Figure 7b). The same marker was associated with stabilizing dough strength under heat stress for lines harboring HMW-GS 2^t^ + 12.1^t^, 5^t^ + 10^t^, and 2^t^ + 10^t^ regardless of the source of the allele.

From the RP1 and RP2 results (Figure 6 and Figure 7), we noticed that the three HMW-GSs (2.1^t^ + 12^t^ 2^t^ + 12.1^t^, and 5^t^ + 10^t^) whether carrying the *Ae. tauschii* alleles or N61 alleles or even heterozygous alleles showed high heat tolerance (RP above 80%) in terms of maintaining high dough strength values under heat stress.

We searched for candidate genes associated with the significant markers. We targeted the markers with a high probability combined with a high R^2^. The resulting candidate genes are listed in Table 3. The markers rs1105119 and rs4262010 on chromosome 2B and 2D, respectively, were associated with dough strength under the conditions at DON19/20 and encoded an MYB transcription factor and Cytochrome P450 protein, respectively (Table 3). The markers rs1201923 on chromosome 4D and rs1240703 on chromosome 6D that were associated with grain yield and dough strength at DON19/20 and HUD19/20, respectively, encoded for glutamine synthase and high-affinity nitrate transporter genes. Under heat-stress environments, most of the markers were associated with enhancing wheat heat-stress tolerance. The marker rs1092278 on chromosome 1D was associated with dough strength under heat-stress encodes for the potassium transporter. Moreover, we found that the marker rs1100384 on chromosome 1D, which was significantly associated with the RP1 of dough strength, encoded protease inhibitor. Both markers rs1055706 on chromosome 4D and rs32025569 on chromosome 6D had a pleiotropic effect on the RP1 and RP2 of dough strength and encoded an NBS-LRR protein and an F-box domain-containing protein, respectively (Table 3). The markers rs1668806|F|0-24 and rs3026863|F|0-12 on chromosomes 4D and 2D were significantly associated with dough strength and grain yield under heat-stress conditions and encoded for protein kinase and pentatricopeptide protein, respectively.

Using the expression data from expVIP databases [61], the expression of the candidate genes was detected and compared to the *Glu-D1* gene expression (*TraesCS1D02G317301*) (Figure 8a,b). The expression of *Glu-D1* was high on seed parts such as endosperm, starchy endosperm, seed coat, and aleurone (Figure 8a). Similarly, the expression level was high during seed developmental stages such as milk and dough developing and repining stages (Figure 8b). The two candidate genes, *TraesCS2B02G387800* on chromosome 2B and *TraesCS4D02G047400* on chromosome 4D, which were associated with dough strength and grain yield under optimum conditions, respectively, showed a high expression on the seed parts and developmental stages similar to the *Glu-D1* gene. Pearson’s correlation indicated a strong association between the expression of these candidate genes (*TraesCS2B02G387800* and *TraesCS4D02G047400*) and *Glu-D1* gene expression (Figure 8a,b). The candidate gene *TraesCS1D02G321000* on chromosome 1D showed high expression during stem elongation and seed germination. The candidate gene *TraesCS1D20G159700* on chromosome 1D showed expression on the floret parts, embryo, and at the booting stage (Figure 8a). We noticed that the candidate gene *TraesCS4D02G136900* on chromosome 4D was expressed during all stages (Figure 8a,b).

## 3. Discussion

This study evaluated the effect of heat stress on flour quality and grain yield under moderate and continuous heat stress in the field using a diverse panel of MSD lines derived from 43 *Ae. tauschii* accessions. The study clearly revealed the significant differential performance of the MSD lines in response to the different thermal gradients used in this study as well as the differences observed among the testing environments for all the different measured characters. We identified the MTAs associated with quality traits and grain yield under heat-stress conditions, as well as heat-stress resilient lines that combined heat-stress tolerance with high grain yield and good end-use quality traits. Our results indicated that the D genome contributed strongly to the grain yield and quality-related characteristics under all conditions, with a diverse range of D-genome markers associated with dough strength and grain yield (Appendix A).

### 3.1. Quality Traits

Previously, we reported a wide variation in the dough strength of MSD lines grown in a cool environment in Japan [17]. In the current study, the MSD lines showed a wide variation in the dough strength under all conditions ranging from optimum to severe heat-stress environments. However, we noticed that the variations in the dough strength of the MSD lines were greater at higher temperatures. This variation reflects the wide genetic diversity in MSD lines in response to heat stress, which has been attributed to various introgression segments of *Ae. tauschii* in the MSD lines [57,58,62]. In our study, most of the significant markers for dough strength were identified on the D-genome confirming the wide diversity of the MSD panel.

The protein content consistently increased with increases in temperature across all environments, especially during the grain-filling period. However, the significantly increased protein content under heat stress was not associated with an increase in dough strength. For instance, the mean protein content was highest in the hottest environment (MED18/19), whereas the mean dough strength was significantly lower than that at MED19/20. Nevertheless, the dough strength at MED19/20 was higher than that at DON19/20 and HUD19/20. This might be due to the fact that some MSD lines maintained high dough-strength values at high temperatures. The increase in the protein content and dough strength, in terms of SDS-SV under heat stress, has been reported earlier under field conditions [20], which might have been due to the increase in both the protein content and protein composition (glutenin and gliadin) in addition to the differences in the growth conditions. We observed that temperatures above 30 °C at MED19/20 led to a significant increase in the dough strength, and a higher temperature at MED18/19 led to a significant decrease in the dough strength. Alvarado et al. [63] reported that the same genotype grown in different environments had the same protein content; however, a significantly weaker dough was observed at the site with the higher mean maximum temperatures. These results signify that not only the protein quantity but also the quality is influenced by heat stress. This can be explained by the fact that gluten components (gliadin and glutenin) do not aggregate synchronously, and thus the gliadin: glutenin ratio is affected. The gliadins are synthesized earlier, whereas glutenins tend to be synthesized later at the grain-filling stage [64,65]. This means that any factor that negatively affects or shortens the grain-filling period can change the gliadin: glutenin ratio and thus negatively affect the dough strength and bread-making quality. In our study, the MSD lines grown at MED18/19 were exposed to an average temperature of 38.5 °C during the grain-filling period, which shortened the grain-filling period. This could be the reason behind the decrease in the dough strength due to the change in the gliadin: glutenin ratio. Although we did not measure the gliadin and glutenin contents directly, their consequent impact on dough strength can be expected.

Previous studies hypothesized that wheat varieties carrying the *Glu-D1d* (5 + 10) allele are largely more tolerant to heat-stress-induced declines in dough quality [4,21,66,67,68]. Our study observed the stable performance of lines possessing subunits 2.1^t^ + 12^t^, 2^t^ + 12.1^t,^ and 5^t^ + 10^t^ derived from *Ae. tauschii*. The dough strengths of the lines carrying these subunits did not differ significantly across the contrasting environments, suggesting that these subunits are associated with heat tolerance.

In this study, two MSD lines (MSD065 and MSD159) showed dough-strength values superior to the recurrent parent (N61) at MED19/20. However, MSD159 was significantly lower than N61 at MED18/19. This could indicate that MSD159 was affected at temperatures above 35 °C. On the other hand, MSD065 maintained better or comparable performance to N61 and the adaptive Sudanese cultivars in terms of dough strength across all environments. Thus, it can be used in breeding programs as a source to improve dough strength even under severe heat-stress conditions.

### 3.2. Grain Yield

The substantial impacts of high temperatures on grain yield were clearly shown across all environments. Temperatures above 35 °C have been reported to be more destructive to wheat grain yield and quality [69,70]. In our study, the high temperatures at MED18/19 led to about a 40% reduction in grain yield compared to the normal conditions at DON19/20, similar to what was reported by Modarresi et al. [39]. In this study, the reduction was significant, even in the two heat-stressed environments (MED18/19 and MED19/20).

Wardlaw et al. [71] stated that the negative impact of temperature increases on grain weight could be interpreted through the various behaviors of carbon (C) and nitrogen (N) metabolisms. High temperatures during grain filling increase the daily flow of C and N through the grain but decrease the flow of C per degree day. Thus, the quantity of C in the grain is more influenced by the temperature than the quantity of N. This explains the decrease in grain yield and the increase in protein content under heat stress.

Despite the significant decrease in grain yield, MSD lines with differential performances across environments were identified. Compared to N61, MSD053 showed significantly higher grain yield at DON19/20 and MED18/19 and was significantly higher than Imam at MED19/20. Two MSD lines (MSD135 and MSD181) gave significantly higher grain yields than N61 at MED18/19 and MED19/20.

### 3.3. Relative Performance under Heat-Stress Conditions

Using the relative performance of each MSD line in the heat-stressed environments, the degree of heat tolerance was estimated for dough strength and grain yield. We identified 75 lines that consistently maintained RP values above 100% for dough strength, which were higher than those of N61 and the adaptive Sudanese cultivar Imam. Likewise, we identified three MSD lines that consistently maintained RP values above 100% for grain yield. By comparing the results of the RPs for dough strength and grain yield, we found that one MSD line (MSD024) that carried HMW-GS 2^t^ + 10^t^ maintained comparable grain yield and dough strength to the recurrent parent with high RP values (above 100). On the other hand, the consistently low RP values of a number of MSD lines indicated the considerable negative effects of high temperatures on their performance in terms of grain yield and dough strength.

Considering the moderate-to-high broad-sense heritability recorded for all the traits studied here, the selections for these traits within the MSD population will be effective.

### 3.4. Marker–Trait Associations for Quality Traits

We could not find stable or pleiotropic MTAs for protein content across the four environments, although the genotype x environment (G × E) interaction was insignificant. All the MTAs identified for the protein content were environment-specific. Consistent with our results, Suliman et al. [47] could not observe stable MTAs for protein content across three environments. We observed the notable contribution of chromosome 6D at DON, where 17 MTAs lying between 365.039Mbp and 471.721Mbp explained 10.5–19% of the phenotypic variances found. Analysis of the gene distribution across A, B, and D genomes revealed the lowest number of gene loci on the D genome compared with the A and B genomes [72]. Leonova et al. [28] detected 50 SNPs across ten chromosomes significantly associated with grain protein content in six environments. Interestingly, some of the MTAs identified on chromosome 5D were very close to an MTA (rs3025015|F|0-20 at 560.305 Mbp) that we identified on chromosome 5D under optimum conditions at DON19/20. Additionally, 1 of the MTAs identified on chromosome 6D (467.9165 Mbp) was very close to a group of 13 MTAs identified on chromosome 6D (457.676–471.721) under optimum conditions at DON19/20. The literature on genetic mapping indicated that QTLs significantly associated with grain protein content were identified in almost all chromosomes of tetraploid and hexaploid wheat [26,27,73]. We could identify MTAs in most of the chromosomes that were previously reported, thereby indicating the wide diversity inherent in the MSD genetic makeup.

In this study, we identified 89 MTAs across 17 chromosomes that were significantly associated with dough strength. Among them, 18 MTAs were stable for dough strength in heat-stressed environments (MED18/19 and MED19/20). These markers could be valuable for the marker-assisted selection of flour quality under heat-stress environments. Interestingly, we noticed that one of the stable markers (rs7352852|F|0-19; 4.891 Mbp) on chromosome 4D that was significantly associated with dough strength under heat-stress conditions was close to the markers rs5332499, rs4440031, and rs3946288, which were identified in MSD lines for hardness under heat and heat–drought stresses [60]. This indicates that this region may contribute to hardness and dough strength under heat-stress conditions.

Although we could not detect stable markers under optimum and heat-stress conditions, a region on chromosome 1D consistently possessed MTAs under these conditions. These results indicate the contribution of other genes in chromosome 1D to dough strength. In this context, we identified different candidate genes associated with dough strength and RP.SSVs under different conditions. The marker rs1105119 on chromosome 2B was associated with dough strength under the optimum conditions at DON19/20 and encoded an MYB transcription factor. The MYB family transcription factors have been reported to play key roles in responses to drought stress and salinity [74,75,76]. Pearson correlation showed a strong association between this candidate gene and the *Glu-D1* gene. Similar to the *Glu-D1* gene, the expression of this gene was on the seed parts at the milk development, dough development, and repining stages. This confirms our GWAS results that identified this marker on chromosome 2B associated with dough strength under the optimum conditions at DON19/20. Liu et al. [29] identified a candidate gene encoded by the MYB transcription factor on chromosome 3B (94.94 cm) associated with protein content in wheat lines derived from wild emmer wheat. The marker *rs4262010* on chromosome 2D, which was associated with dough strength under the optimum conditions at DON19/20, encoded a cytochrome P450 protein, which is the enzyme that can perform several types of oxidation-reduction reactions. Candidate genes encoding cytochrome P450 on chromosomes 2A, 2B, 5A, and 7B that were shown to be associated with protein content have been identified [29]. Likewise, the wheat cytochrome P450 was found to enhance resistance to deoxynivalenol and grain yield [77]. Although the identified candidate gene was associated with dough strength under optimum conditions and not with grain yield, a marker (3024386|F|0-37) identified for grain yield in the same environment was very close (599.99 Mbp) to this MTA on the same chromosome (Appendix A). This confirms the association of cytochrome P450 with grain yield and reveals its association with dough strength. However, the expression of this gene was on floret, rachis, and spikelets, and was not close to the *Glu-D1* gene (Figure 8a,b).

Under the heat-stress conditions at MED19/20, the marker rs1092278|F|0-29 on chromosome 1D was associated with dough strength and found to encode a potassium transporter, which is reported to play an essential role in plant growth and environmental adaptation and regulate potassium uptake in wheat [78]. It has been reported that the HMW-GS levels were regulated by foliar spraying of potassium fertilizer [79]. Furthermore, the marker rs1055706|F|0-65 on chromosome 4D that was stable for dough strength under heat-stress conditions and controlled both the grain yield and RP2.SSVs under the heat-stress conditions at MED18/19 and MED19/20, respectively, was found to be associated with the candidate gene *TraesCS4D02G136900* that encodes NBS-LRR. The TaRPM1 is a type of CC-NBS-LRR that positively regulates wheat response to high temperatures. In wheat, Wang et al. [80] concluded that TaRPM1 positively regulates the high-temperature seedling-plant (HTSP) resistance to *Puccinia striiformis* f. sp. *tritici* (*Pst*) through the salicylic acid (SA) signaling pathway. These results could suggest that factors controlling dough strength stability are associated with plant defense against pathogens.

The marker rs32025569|F|0-33 on chromosome 6D was associated with RP1.SSVs and RP2.SSVs. The candidate gene for this marker encodes F-box proteins that play crucial roles in abiotic stress responses and have been reported to enhance heat-stress tolerance in wheat by improving enzymatic antioxidants [81]. Thus, these results indicated their contribution to heat-stress tolerance in wheat and dough-strength stability. On the other hand, a study on the MSD panel identified F-box proteins on chromosome 6A associated with kernel weight and kernel diameter under optimum conditions [59].

In addition to the identification of candidate genes, we identified markers that controlled more than one trait under different conditions. Chromosome 4D showed the highest contribution of markers with pleiotropic effects. Similar to our results, a previous study involving MSD lines showed chromosome 4D with the highest contribution of MTAs identified for hardness under optimum, heat, and heat–drought conditions, as well as the hardness heat index and hardness heat drought index [60]. Interestingly, the pleiotropic marker rs998809|F|0-7 that controlled both the grain yield and dough strength under heat stress at MED18/19, overlapped with the marker rs1043872|F|0-49 that was identified for hardness under heat-stress conditions in MSD lines [60]. These results indicate that the region on chromosome 4D harbors MTAs that control important quality traits and grain yield under heat-stress conditions. Therefore, this region could be used in marker-assisted selection targeting these traits under heat-stress conditions. Moreover, we found the pleiotropic marker 1079306|F|0-62 on chromosome 4D that controlled grain yield under optimum conditions and RP1.SSVs. Similarly, Itam et al. (2021b) [58] found that the same marker (1079306|F|0-62) on chromosome 4D controls plant height under heat and combined heat–drought stress in MSD lines.

We found MTAs with a pleiotropic effect on grain yield at DON19/20 and dough strength at MED18/19, as well as RP1.SSVs. Likewise, we found MTAs that controlled dough strength at MED18/19 and MED19/20, grain yield at MED18/19 and DON19/20, as well as RP1.SSVs, RP2.SSVs, RP1.GY, and RP2.GY. These MTAs could be utilized for marker-assisted selection targeting flour quality and grain yield and their stability under heat stress. To the best of our knowledge, this is the first report of MTAs controlling dough strength and grain yield under heat-stress conditions. In our previous study involving the MSD lines [17], we reported no negative relationship between the dough strength and grain yield under optimum conditions, indicating the suitability of the MSD in breeding high dough strengths without a negative effect on grain yield. In our study, most of the markers identified for the RP and pleiotropic markers were on chromosome 4D, confirming the contribution of this chromosome to heat-stress tolerance. Thus, collaborative work/research of gene mining on chromosome 4D would facilitate the production of cultivars that combine different desired traits.

The few common MTAs for dough strength and grain yield may be due to the complexity of the MSD population, which has a huge diversity resulting from the diverse D-genome sources. After validation, these markers could be used in wheat molecular breeding for the identified traits under optimum and heat-stress conditions.

Our results clearly showed the association of the different candidate genes with the identified markers for dough strength under heat and optimum conditions. Therefore, since most of the identified markers for dough strength were under heat stress, this may indicate that other genetic factors contribute to dough strength, especially under continuous heat-stress environments. Thus, the cooperative expression of these MTAs under both heat-stress and optimum conditions may contribute to wheat’s quality stability and heat-stress tolerance.

### 3.5. Marker–Trait Associations for Grain Yield

Concerning grain yield and related traits, the D genome has been reported to possess the lowest number of loci, consistent with its relatively lowest diversity [42,82]. In our study, we identified 53 significant MTAs across all environments, with the highest contribution shown by MTAs on the D genome under optimum and heat-stress conditions. This indicates that the diversity of hexaploid wheat was successfully increased by the introgression of *Ae. tauschii’s* D genome.

Previous studies identified MTAs on chromosomes 5A, 6A, 3B, and 5B for grain yield in temperate and heat-stress environments [45,47]. Moreover, Li et al. [42] identified QTLs for grain yield on chromosomes 2D, 3D, and 5A. These markers overlapped with the markers identified here for grain yield on chromosomes 2D and 3D under optimum and heat-stress conditions and chromosome 5A under optimum conditions. We found that the marker rs1201923|F|0-5 on chromosome 4D contributed to grain yield at DON19/20 (R^2^ = 0.20). The associated candidate gene *TraesCS4D02G047400* encoded glutamine synthetase, which regulates nitrogen metabolism in wheat [83]. Glutamine synthetase has been reported to play an essential role in nitrogen-use efficiency, uptake, and assimilation [84]. These findings indicate that these MTAs were associated with nitrogen-use efficiency genes in the optimum environment. Interestingly, based on MSD lines, Elhadi et al. [60] identified the same candidate gene (*TraesCS4D02G047400*) on chromosome 4D, which was associated with hardness under heat and combined heat–drought and hardness indexes. These results indicate a potential association between genes underlying hardness and grain yield, as well as with nitrogen-use efficiency genes in MSD lines under optimum and heat-stress conditions. Although this candidate gene was associated with grain yield under optimum conditions, its expression was similar to the *Glu-D1* gene, which might suggest its potential contribution to dough strength. The position of this marker (encoding glutamine synthetase) lay on the same (exact) region (23.837 Mbp) of the marker (1201923|F|0-38) that was identified for dough strength under heat stress (MED18/19). Thus, this may explain its similar expression to the *Glu-D1* gene.

Under heat-stress conditions (MED18/18), the marker rs3026863|F|0-12 on chromosome 2D was found to control the grain yield. The candidate genes of this marker encoded pentatricopeptide (PPR) proteins, which have been reported to be important in regulating plant growth, development, cytoplasmic male sterility, stress responses, and seed development [85]. The expression of this gene was not close to *Glu-D1*.

Stable MTAs could not be spotted for grain yield across the four environments, indicating that the trait was significantly influenced by the environment and genotype x environment (G × E) interaction.

Generally, the identified MTAs in this study can be used to understand the genetic responses to heat stress regarding the quality traits and grain yield, thus facilitating the introduction of desirable and stable alleles to develop resilient cultivars that combine both grain yield and end-use quality under heat stress using marker-assisted selection.

### 3.6. Alleles’ Contribution

The alleles’ contribution of markers that were associated with heat tolerance and stability of dough strength showed that both the N61 and *Ae. tauschii* alleles contributed either negatively or positively to the dough strength stability in each subunit. The absence of a clear relationship between the stability of dough strength in the lines with subunits 2.1^t^ + 12^t^, 2^t^ + 12.1^t^, and 5^t^ + 10^t^ can be attributed genetically to the small number of lines carrying these subunits. Thus, more investigation is needed. However, we observed that irrespective of whether the allele originated from N61 or *Ae. tauschii*, the three subunits 2.1^t^ + 12^t^, 2^t^ + 12.1^t^, and 5^t^ + 10^t^ showed high RPs above 80%, which indicates that their heat tolerance or stability could be mainly due to their subunits at the *Glu-D1* locus and not due to the identified alleles (N61 or *Ae. tauschii*). This observation is consistent with our phenotypic results.

The results of this study demonstrated the significant effects of integrating the D genome from diverse *Ae. tauschii* accessions into a beard wheat genome. This is positively reflected in the identification of MSD lines that showed a good and stable grain yield as well as good quality-related characteristics under moderate and severe heat-stress environments.

## 4. Materials and Methods

### 4.1. Plant Materials

This study used a multiple synthetic derivative (MSD) panel that was developed by crossing and backcrossing the Japanese common wheat cultivar ‘Norin 61’ (hereafter referred to as N61) with 43 synthetic hexaploid wheat (SHW) lines [54,55]. The 43 SHW lines were developed by crosses between 43 diverse accessions of *Ae. tauschii* and *T. turgidum* var. durum cv. ‘Langdon’ (LDN) [86,87]. The experiment consisted of 147 MSD lines (BC_1_F_6_ in the 2018/19 season and BC_1_F_7_ in the 2019/20 season) in addition to three check cultivars. The three check cultivars included the recurrent parent, N61, and two adapted Sudanese cultivars (Imam and Goumria).

### 4.2. The Experimental Sites and Field Management

The study was carried out in four environments located at three agro-ecological sites in Sudan (Appendix A): DON, located in the Northern State (19°08′ N, 30°27′ E, 239 masl), HUD, located in the River Nile State (17°35′ N, 33°50′ E, 409 masl), MED at the Gezira Research Farm, Agricultural Research Corporation, in the central clay plain of Gezira State (14°24′ N, 29°33′ E, 407 masl). The soil texture at DON is sandy clay loam at 0–30 cm and silty clay loam at 30–60 cm with a pH of 8.0 and low organic matter content (<5%). The soil of HUD is classified as a middle-terrace soil (Karu; pH 8), whereas the soil of MED is a heavy clay soil (pH 8.0–8.4) with low organic matter content (<5%) and low levels of nitrogen (380 ppm) and phosphorus. The experiments were conducted during the 2018/2019 season at MED (MED18/19) and the 2019/2020 season at MED (MED19/20), HUD (HUD19/20), and DON (DON19/20). The Gezira Research Farm at MED has been classified as a mega-environment 5B (ME5B) for wheat cultivation [88]. The characteristics of each environment have been described in Elbashir et al. [56]. We considered DON the optimum environment for wheat cultivation in Sudan, whereas HUD and MED were considered moderate and continuous heat-stressed environments, respectively. All experiments in the four environments were arranged in an alpha lattice design with two replications. Each line was sown in a plot consisting of four rows, 1 m-long and 0.2 m apart. Thus, the harvested area was 4 rows × 1 m long × 0.2 m between rows. At MED18/19, MED19/20, and HUD19/20, the sowing was carried out during the 4th week of November, whereas at DON19/20, the sowing was on the 3rd of December 2019.

The seeds were treated with the insecticide Gaucho (imidacloprid, 35% WP, Bayer Crop Science, Kansas City, MO, USA) and the fungicide Raxil (tebuconazole) at 0.75 and 1.25 g/kg of seed, respectively, to control termites, aphids, and soil-borne diseases. The treated seeds were manually sown at a rate of 120 kg ha^–1^. Superphosphate was applied by furrow placement before sowing at a rate of 43 kg ha^–1^ of P_2_O_5_. Two doses of nitrogen (86 kg N ha^–1^) were applied in the form of urea; the first dose was at the three-leaf stage (second irrigation) and the second dose was at the tillering stage (fourth irrigation). The experiments were irrigated frequently every 10–12 days to avoid exposure to water stress. Hand weeding was performed at least twice to keep the field free of weed infestation. It is worth mentioning that no serious diseases were reported in the three agro-ecological sites where the experiments were conducted.

Daily maximum and minimum temperature data during the two cropping seasons (2018/2019 and 2019/20) for the three agro-ecological sites were kindly provided by the Sudan Meteorological Authority.

### 4.3. Evaluation of Flour Quality

Five grams from each MSD line were milled using a UDY cyclone sample mill (UDY Corp., Fort Collins, CO, USA) equipped with a 1 mm screen to obtain whole wheat flour. The protein content was measured as a percentage of the total weight by near-infrared spectroscopy (NIR composition analyzer KJT-270, Kett Electric Laboratory Co., Ltd., Ota, Tokyo, Japan). The SDS sedimentation volume (SDS-SV) was measured using the method of Takata et al. [89] to assess the gluten quantity and quality. As the sedimentation volume is highly correlated with dough strength and bread loaf volume [90], the specific sedimentation values (which are highly correlated with dough strength) were calculated as an index of the gluten quality, by dividing the SDS sedimentation volume (mL) by the protein content (%). The protein content has also been reported to be highly correlated with sedimentation volume [91,92].

### 4.4. Genome-Wide Association Analysis (GWAS)

We performed GWAS using DArT-seq markers (Diversity Arrays Technology, Bruce, Australia “https://www.diversityarrays.com (accessed on 9 February 2021)” for 127 MSD lines and N61. A mixed linear model (MLM) was adopted, including the population structure and kinship matrix using TASSEL v. 5.2.66 software [93]. A total of 19,155 high-quality SNP markers with a call rate of 90% (10% missing data) and MAF (minor allele frequency) of >0.05 were used in the analysis. Manhattan plots were created using (−log10) (P). The adjusted threshold of *p* < 3 × 10^−3^ was used to refer to the degree of association between each SNP marker and a trait, whereas R^2^ referred to the variation explained by the significantly associated markers. To draw the Manhattan plots and quantile–quantile plots, we used the MLM product from TASSEL in R v. 4.0.3 with custom scripts in the developed GWAS package rMVP [94].

### 4.5. Candidate Genes and Gene Expression

To identify the candidate genes for the dough strength, grain yield, and relative performance indices, we selected the top MTAs that were identified for each trait and BLAST them against the International Wheat Genome Sequencing Consortium (IWGS) RefSeq V.1 chromosomes, using URGI with the BLAST option “https://urgi.versailles.inra.fr/blast/ (accessed on 2 June 2022)”. Then, we searched for the candidate genes with high confidence in the distance (±500 kbp) for the genome region. We used version 2.1 of IWGSC_Ref_seq to search for genes with high confidence. We used version 1.1 of IWGSC_Ref_Seq_Annotations along with EnsemblPlant “https://plants.ensembl.org (accessed on 5 June 2022)” to identify the protein function.

We investigated the expression levels of all candidate genes that highly contributed to dough strength and compared them to the expression of the *Glu-D1* genes using the Wheat Expression Browser expVIP. This led to understanding the association between the candidate genes and dough strength.

### 4.6. Statistical Analysis

Phenotypic data were subjected to analysis of variance separately for each environment and then combined analyses were carried out. A total of 129 MSD lines, the data of which were commonly available in the four environments, were analyzed using the GenStat Software (18th edition). We used the least significant difference (LSD, 0.05) for genotype mean separation and the Tukey test to compare the mean of each trait across all environments using SPSS software (version 25.0.1). Broad-sense heritability (H2) was calculated using Plant Breeding Tools v. 1.4 software (International Rice Research Institute, http://bbi.irri.org/products (accessed on 6 June 2022)).

### 4.7. Relative Performance

To compare the performance of the MSD lines under heat-stress conditions relative to the optimum environment, the relative performance (RP) was calculated considering DON19/20 as an unstressed (optimum) environment and both MED18/19 and MED19/20 as heat-stressed environments. The RP values for dough strength and grain yield for each line were calculated as
$$\frac{\mathrm{Phenotypic}\;\mathrm{value}\;\mathrm{of}\;\mathrm{each}\;\mathrm{line}\;\mathrm{under}\;\mathrm{heat}\;\mathrm{stress}\;\mathrm{environment}\;\times \;100}{\mathrm{Phenotypic}\;\mathrm{value}\;\mathrm{of}\;\mathrm{each}\;\mathrm{line}\;\mathrm{under}\;\mathrm{optimum}\;\mathrm{environment}}$$

Two RP values were calculated for each line, one for MED18/19 (RP1) and the other for MED19/20 (RP2).

## 5. Conclusions

With the expectation that *Ae. tauschii* genes could enhance wheat-bread-making quality characteristics under heat-stress conditions, in this study we explored the variations in dough strength and grain yield in a diverse population of MSD lines (harboring different *Ae. tauschii* introgressions) and conducted GWAS. We found considerable genetic variations for both traits and identified several MTAs, most of them on the D genome, under optimum, moderate, and continuous heat-stress conditions. We identified one MSD line (MSD024) that maintained comparable grain yield and dough strength to the recurrent parent with high heat-tolerance efficiency. We found that the presence of three HMW-GS alleles at the *Glu-D1* locus (2.1^t^ + 12^t^, 2^t^ + 12.1^t^, and 5^t^ + 10^t^) derived from *Ae. tauschii* was significantly associated with relatively stable dough strength across the four environments, which ranged from optimum to severe heat-stressed conditions. These alleles could be used for future improvements of wheat’s end-use quality characteristics under severe heat stress. We successfully identified several chromosomal regions affecting grain yield and dough strength, representing a potential target for MAS to improve both traits under optimum and heat-stress conditions. We documented that chromosome 4D in MSD lines harbors promising regions/genes that control different traits under different conditions. Thus, after the validation of these MTAs, the collaborative work/research of gene mining on chromosome 4D would facilitate the production of cultivars that combine desired traits. In addition, we identified several candidate genes associated with dough strength and grain yield. This study represents one of the rare cases where a large population has been studied for the grain yield and quality traits under field conditions with temperature gradients ranging from relatively optimum to moderate and continuous heat stress. The study provided valuable germplasm lines and potential markers, which are useful for further applications in wheat molecular breeding. Moreover, our results emphasized the importance of *Ae. tauschii* as a great genetic resource for wheat productivity and flour quality improvement in the face of increasing climate change.

## Figures and Tables

**Figure 1 ijms-23-12034-f001:**
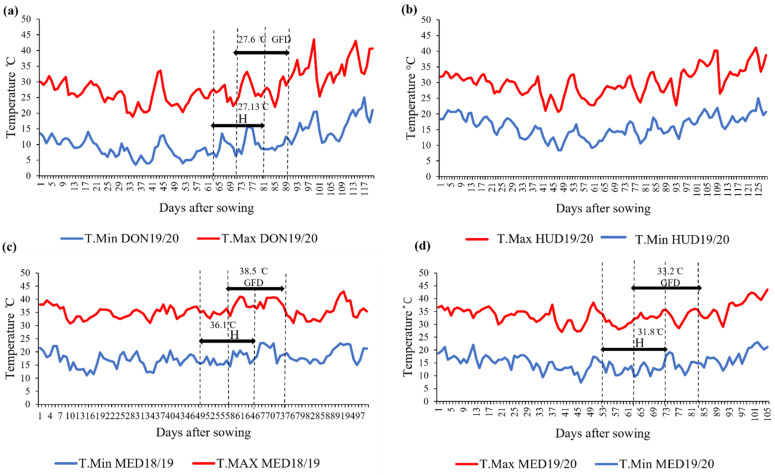
Daily maximum and minimum temperatures (T.Max and T.Min, respectively) during the wheat-cropping season at (**a**) DON19/20, (**b**) HUD19/20, (**c**) MED18/19, and (**d**) MED19/20. H: heading stage; GFD: grain-filling duration. The average maximum temperatures during the heading and grain-filling stages are shown at each location, except for HUD19/20 (no data).

**Figure 2 ijms-23-12034-f002:**
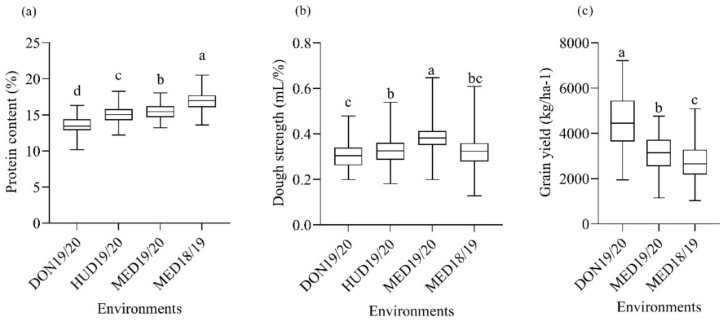
Box plot of (**a**) mean protein content, (**b**) dough strength, and (**c**) grain yield of multiple synthetic derivative lines grown under optimum (DON19/20), moderate heat-stress (HUD19/20), and continuous heat-stress (MED18/19 and MED19/2020) conditions. Box plots with similar lower-case letters indicate that the means for the trait are not significantly different according to the Tukey test at *p* < 0.05.

**Figure 3 ijms-23-12034-f003:**
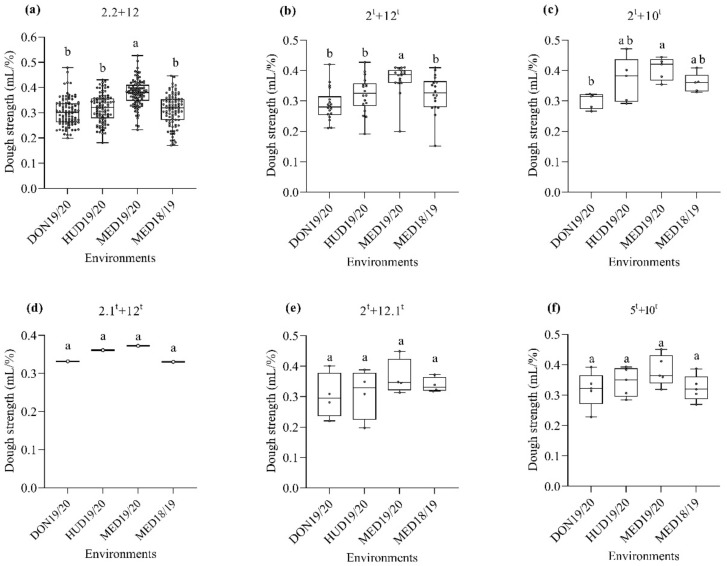
Box plot for the dough strength (mL/%) as affected by the HMW-GS at the *Glu-D1* locus across the four environments. (**a**), subunits 2.2 + 12; (**b**), subunit 2^t^ + 12^t^; (**c**), subunit 2^t^ + 10^t^; (**d**), subunit 2.1^t^ + 12^t^; (**e**), subunit 2^t^ + 12.1^t^; (**f**), subunit 5^t^ + 10^t^. Similar lower-case letters indicate that the means for the HMW-GS pairs are not significantly different according to the Tukey test at *p* < 0.05.

**Figure 4 ijms-23-12034-f004:**
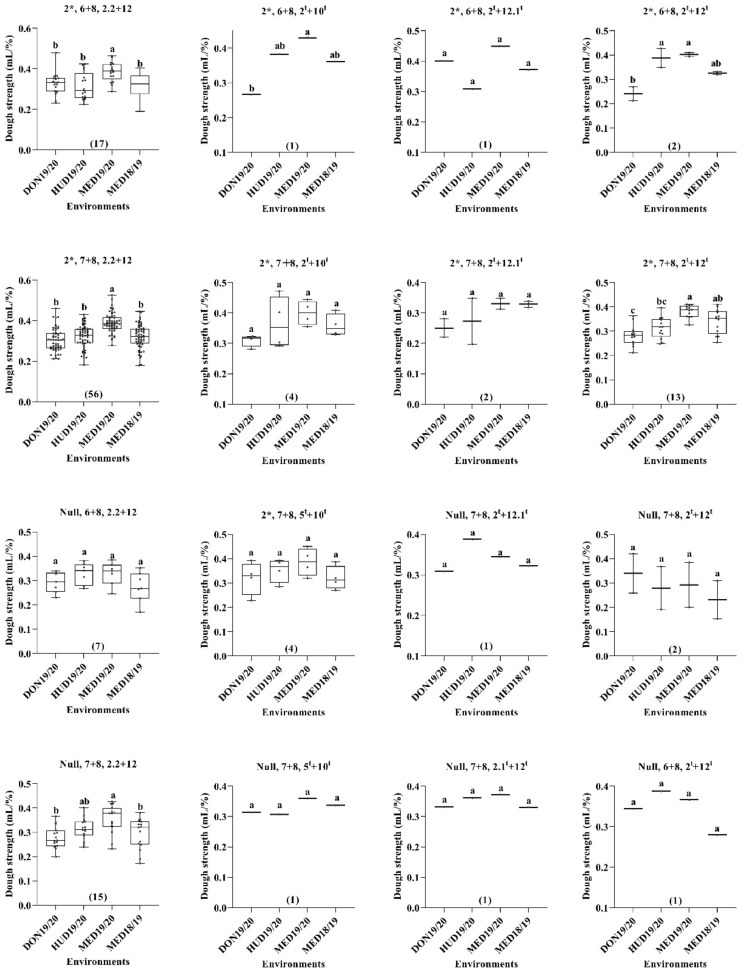
Box plot for the dough strength (mL/%) as affected by the HMW-GS combination across the four environments. Numbers in parentheses indicate the number of lines. Similar lower-case letters indicate that the means for the HMW-GS pairs are not significantly different according to the Tukey test at *p* < 0.05.

**Figure 5 ijms-23-12034-f005:**
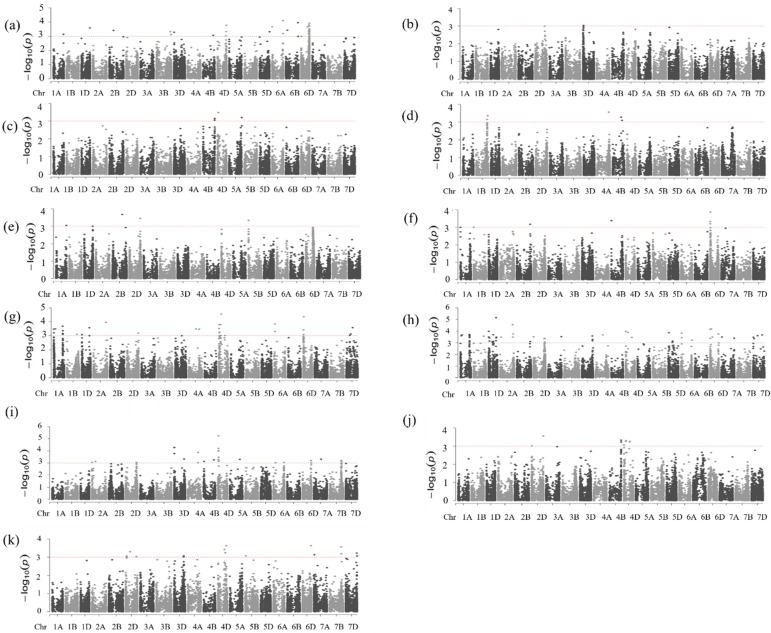
Manhattan plots for protein content (**a**–**d**) at DON19/29, HUD19/20, MED18/19, and MED19/20, respectively, for dough strength, (**e**–**h**), at DON19/29, HUD19/20, MED18/19 and MED19/20, respectively, and for grain yield (**i**–**k**) at DON19/20, MED18/19, and MED19/20, respectively.

**Figure 6 ijms-23-12034-f006:**
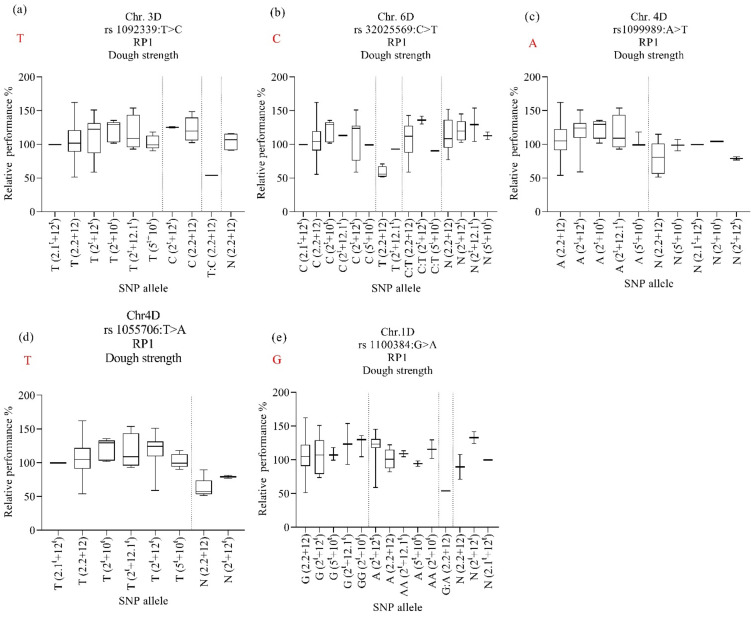
Effect of selected marker–trait associations (**a**–**e**) on relative performance of dough strength (PR1) in lines with different HMW-GS in MSD lines grown under heat stress conditions. A, adenine; C, cytosine; T, thymine; G, guanine; N, unknown. Alleles in red refer to those of the backcross parent of the population, ‘Norin 61’.

**Figure 7 ijms-23-12034-f007:**
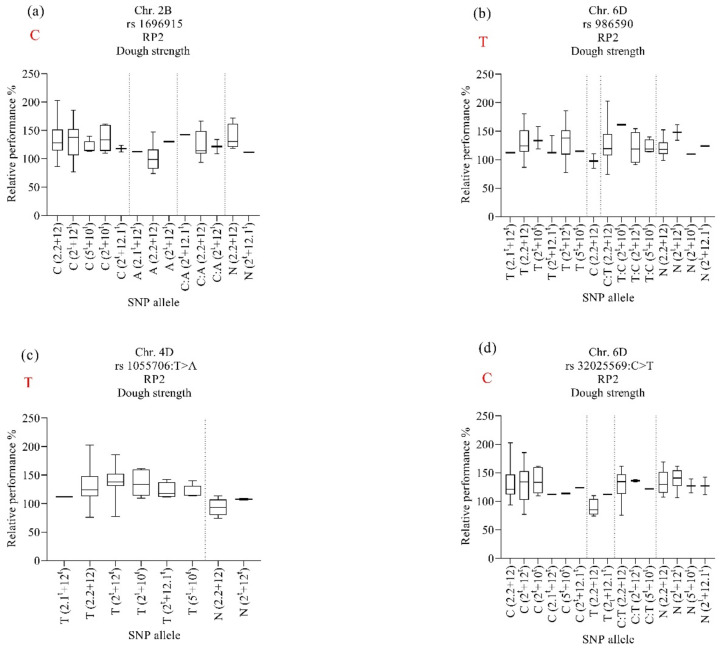
Effect of selected marker–trait associations (**a**–**d**) on relative performance of dough strength (PR2) in lines with different HMW-GS in MSD lines grown under heat stress. A, adenine; C, cytosine; T, thymine; G, guanine; N, unknown. Alleles in red refer to those of the backcross parent of the population, ‘Norin 61’.

**Figure 8 ijms-23-12034-f008:**
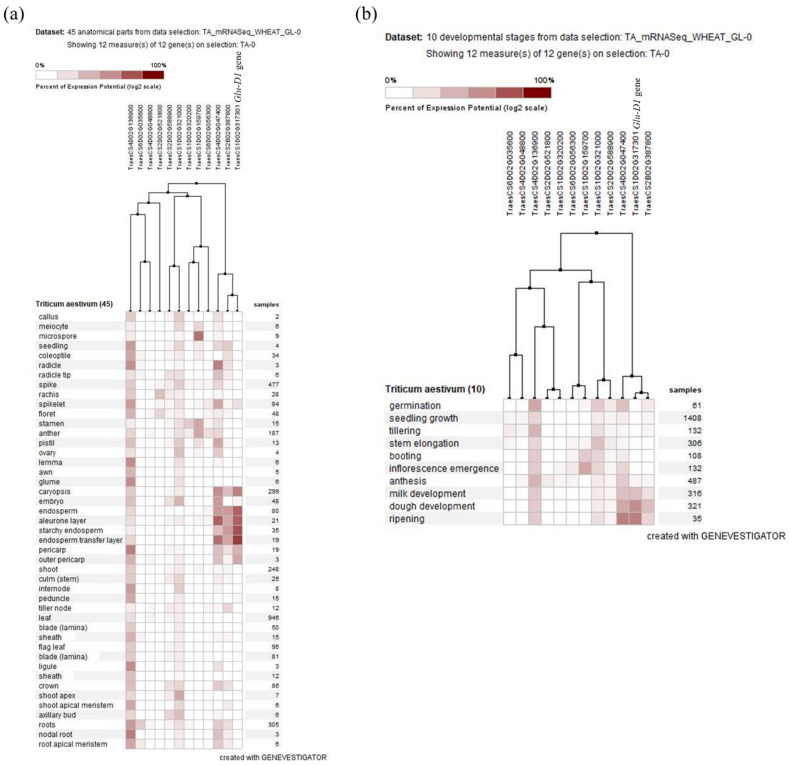
The expression of the candidate genes compared to the *Glu-D1* gene expression (*TraesCS1D02G317301*) in different anatomical parts (**a**) and at different developmental stages (**b**).

**Table 1 ijms-23-12034-t001:** Effects of genotype (G), environment (E), and their interaction on dough strength, protein content, and grain yield of multiple synthetic derivative lines grown under optimum (DON19/20) and heat-stress (HUD19/20, MED18/19, and MED19/2020) conditions.

Protein Content (%)						Dough Strength (mL/%)					Grain Yield kg/ha^−1^			
	DON19/20	HUD19/20	MED18/19	MED19/20	C	DON19/20	HUD19/20	MED18/19	MED19/20	C	DON19/20	MED18/19	MED19/20	C
G	<0.001	0.103	<0.001	<0.001	<0.001	0.003	<0.001	<0.001	<0.001	<0.001	<0.001	<0.001	<0.001	<0.001
E	-	-		-	<0.001	-	-	-	-	<0.001	-	-	-	<0.001
G × E	-	-		-	0.56	-	-	-	-	<0.001	-	-	-	<0.001
Mean	13.6	15.1	17	15.5	15.5	0.30	0.32	0.32	0.38	0.33	4485	2733	3133	3433
Range	10.2–16.3	12.2–18.3	14.4–20.5	13.2–18.1	13.8–17.6	0.20–0.48	0.18–0.47	0.15–0.45	0.20–0.53	0.20–0.44	1942–7218	1030–5084	1150–4764	2154–5395
CV%	4.2	7.4	4.9	3.5	8.5	15.5	12.7	8.9	10.0	14.0	21.34	22.21	18.76	25.2
LSD	1.25	2.58	1.63	1.12	2.85	0.104	0.093	0.056	0.079	0.101	1850	1175.4	1132	1711
SE±	0.57	1.12	0.83	0.54	1.31	0.047	0.041	0.028	0.038	0.046	934.5	593.8	571.9	870.4
h^2^	-	-	-	-	0.68	-	-	-	-	0.86	-	-	-	0.58
Goumria	14.6	15.8	17.0	16.1	15.9	0.33	0.37	0.42	0.42	0.39	6438	2188	4166	4258
Imam	12.9	15.6	14.6	13.2	14.2	0.39	0.35	0.45	0.47	0.41	6969	3594	4387	4965
Norin 61	13.5	16.4	15.5	14.8	15.1	0.41	0.38	0.39	0.43	0.40	4579	2636	3277	3494

C, combined analysis; G, the main genotype effect in each environment; E, environment main effect; G × E, genotype-by-environment interaction; h^2^, heritability estimate; SE±, standard error of differences; LSD, least significant difference; CV%, coefficient of variation.

**Table 2 ijms-23-12034-t002:** Stable and pleiotropic MTAs of dough strength (SSVs), relative performance of dough strength (RP.SSVs), grain yield (GY), and relative performance of grain yield (RP.GY) in multiple synthetic derivative lines grown under optimum (DON19/20) and heat (HUD19/20, MED18/19, and MED19/20) conditions.

			Environments				
Marker	Chr	Pos	DON19/20	HUD19/20	MED18/19	MED19/20	R^2^
1204551|F|0-57	1A	500.253			(SSVs)	(SSVs)	11.2–14.0
1094315|F|0-45	1A	506.846			(SSVs)	(SSVs)	9.8–11.6
3959168|F|0-15	1A	508.265			(SSVs)	(SSVs)	9.8–11.6
1210578|F|0-9	1A	510.293			(SSVs)	(SSVs)	9.9–12.0
3947627|F|0-33	1A	510.911			(SSVs)	(SSVs)	12.7–14.1
4910833|F|0-62	1A	511.070			(SSVs)	(SSVs)	9.8–11.6
2303774|F|0-6	1A	513.884			(SSVs)	(SSVs)	10.9–12.1
3953635|F|0-16	1A	97.919			(SSVs)	(SSVs)	15.1–17.3
996849|F|0-11	1B	559.059			(SSVs)	(SSVs)	9.8–11.6
1092278|F|0-29	1D	412.338			(SSVs)	(SSVs)	12.7–22.3
1696345|F|0-38	1D	421.872			(SSVs)	(SSVs)	12.1–13.8
994055|F|0-66	2A	697.354			(SSVs)	(SSVs)	17.2–20.2
1088439|F|0-52	3D	14.283	(GY)			(RP2.GY)	13.2–15.4
1042486|F|0-52	4A	577.563	(GY)		(SSVs), (RP1.SSVs)		13.1–15.5
3953635|F|0-25	4A	403.756			(SSVs)	(SSVs)	15.1–17.3
1134011|F|0-55	4B	585.751	(GY)		(RP1.GY)		12.0–13.1
1201923|F|0-38	4D	23.837	(GY)		(SSVs)		10.5–18.3
1062681|F|0-26	4D	23.470	(GY)		(SSVs)		13.1–17.6
1051116|F|0-23	4D	335.228			(SSVs), (GY), (RP1.SSVs)		9.4–10.2
1055706|F|0-65	4D	123.018			(SSVs), (GY)	(SSVs), (RP2.SSVs)	13.2–18.9
7352852|F|0-19	4D	4.891			(SSVs)	(SSVs)	10.0–13.6
1079306|F|0-62	4D	25.702	(GY)		(RP1.SSVs)		13.1
998809|F|0-7	4D	98.475			(SSVs), (GY)		9.5–10.2
3534425|F|0-23	6A	37.593			(SSVs)	(SSVs)	10.4–13.9
3940208|F|0-6	6A	39.432			(SSVs)	(SSVs)	12.2–12.3
1091824|F|0-36	7D	245.790			(SSVs)	(SSVs)	9.9–11.8

Ch, chromosome; Pos, Position; R^2^, the variation explained by the significantly associated markers.

**Table 3 ijms-23-12034-t003:** Candidate genes for strong marker–trait associations identified for grain yield, dough strength, and relative performance calculated for dough strength under different environments.

Marker	Environment	Trait	Ch.	R^2^	Gene	Protein	Function
1105119|F|0-22	DON19/20	SSVs	2B	0.15	*TraesCS2B02G387800*	MYB transcription factor	Drought stress response in wheat
4262010|F|0-9	DON19/20	SSVs	2D	0.14	*TraesCS2D02G521800*	Cytochrome P450 family protein	Enhanced biotic stress resistance and grain development in wheat
1201923|F|0-5	DON19/20	GY	4D	0.20	*TraesCS4D02G047400*	Glutamine synthase	Regulated nitrogen metabolism in wheat
1240703|F|0-26	HUD19/20	SSVs	6D	0.13	*TraesCS6D02G035600*	High-affinity nitrate transporter	Improved nitrogen uptake, root growth, and grain yield in wheat
1668806|F|0-24	MED18/19	SSVs	4D	0.19	*TraesCS4D02G048800*	Protein kinase	Regulated plant development and stress tolerance in wheat
3026863|F|0-12	MED18/19	GY	2D	0.14	*TraesCS2D02G588900*	Pentatricopeptide repeat-containing family protein	Regulated plant growth, development, cytoplasmic male sterility, stress responses, and seed development
1092278|F|0-29	MED19/20 MED18/19	SSVs	1D	0.22	*TraesCS1D02G321000*	F-box family protein	Enhanced tolerance to oxidative stress in wheat
					*TraesCS1D02G320200*	Potassium transporter	Regulated potassium uptake in wheat. HMW-GL levels were regulated by potassium availability
1055706|F|0-65	MED18/19	SSVs GY	4D	0.18	*TraesCS4D02G136900*	NBS-LRR disease resistance protein, putative, expressed	*TaRPM1* is a type of CC-NBS-LRR that positively regulated wheat at high temperatures
	MED19/20	SSVs RP2.SSVs					
32025569|F|0-33		RP1.SSVs RP2.SSVs	6D	0.19	*TraesCS6D02G056300*	F-box and associated interaction domains-containing protein TE	Enhance heat-stress tolerance in wheat through improve enzymatic antioxidant
1100384|F|0-67		RP1.SSVs	1D	0.13	*TraesCS1D02G159700*	Protease inhibitor/seed storage/lipid transfer family protein	Seed storage protein regulated elasticity and extensibility of dough that determine the processing qualities of various end-products

SSVs, specific sedimentation values (dough strength); RP.SSVs, relative performance for dough strength; GY, grain yield; Ch.; chromosome.

## Data Availability

Not applicable.

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
