# Peer review of "Identification of Glu-D1 Alleles and Novel Marker–Trait Associations for Flour Quality and Grain Yield Traits under Heat-Stress Environments in Wheat Lines Derived from Diverse Accessions of Aegilops tauschii"

_ijms, 2022, doi:10.3390/ijms231912034_

Round 1
Reviewer 1 Report
The study and the manuscript are well prepared and represent excellent contribution to our knowledge on grain quality and yield in relation to use of synthetic wheat under effect of different environment. The study strength is the use of GWAS as an instrument to learn about genetic control of important traits rather than the goal itself. The methodology is relevant and justified. However, as always, the manuscript can be improved.
1. Short information on wheat production and breeding in Sudan is well justified.
2. Trial management. The authors did not specify if the fungicides were applied to protect the plots against diseases or any diseases were observed which can substantially affect the grain yield and protein content.
3. Section 2.2 on dough strength focuses on relative performance of Glu subunits and their combinations across four environments. However, it misses their comparison. Does it make sense to compare the average values of alternative subunits for A, B and D genome independently without considering the variation for two other genomes. If the number of individuals with each alternative subunit is comparable – it is probably worth doing.
4. Figure 1 b does not have the average Tmin and Tmax values.
5. Tables 2 and 3. The same markers are presented in different format.
6. The authors argue that the main difference at three experimental sites is temperature. However, it may not be true and other factors could be involved at cites as remote from each other as 900 km. This includes soil, climate, etc. The explanation in this respect and perhaps reference to previous multilocational studies would be useful. By the way, a map with the trials sites and wheat production regions in the text or supplement will be useful for readers understanding of the country wheat geography and in promotion of Sudan as strong agricultural player.
7. Supplementary tables 1 and 2 could be combined.
8. The discussion section seems to be too long again with reference to figures and tables. This can be better focused on comparison with similar studies and a broader meaning of the findings for research and breeding activities/strategies.
9. The manuscript needs careful reading once again to avoid the following mistakes “..glutenin subunits (HMW-GSs) of the glutenin..” , “..heat tolerant/stability..”.
Reviewer 2 Report
The manuscript entitled; “Identification of Glu-D1 alleles and novel marker-trait associations for flour quality and grain yield traits under heat stress environments in wheat lines derived from diverse accessions of Aegilops tauschii” by Mohamed et al. describes about the role of D-genome to improve heat stress as well as quality of 147 bread wheat lines. Overall, manuscript is written well, but I have few concerns regarding data analysis and presentation of results. Further, I will suggest following changes before its acceptance.
Abstract
Please indicate the number of lines and markers used in the current experiment
A clear take home message is missing in the concluding paragraph of the manuscript
Introduction
Line 104, please add the reference” Shokat S, Novák O, Široká J, Singh S, Gill KS, Roitsch T, Großkinsky DK, Liu F. (2021). Elevated CO2 modulates the effect of heat stress responses in Triticum aestivum by differential expression of isoflavone reductase-like (IRL) gene. Journal Experimental Botany, 72: 7594–7609.
Line 118, pleas add reference “Singh et al (2018). Harnessing genetic potential of wheat germplasm banks through impact-oriented-prebreeding for future food and nutritional security. Scientific Reports, 8: 12527.
Please indicate a concrete hypothesis in the concluding paragraph to indicate how the sub-objectives were achieved.
Results
I think there are three factors i.e., genotypes, stress conditions and years and in my opinion there should be three-way analysis of variance. Results should be rewritten after doing the three-way analysis of variance.
Overall, results should be written concisely. Current presentation does not make sense and can’t be eye catching. Please rewrite to express the meanings. For example, results of analysis of variance can be written as follow, grain yield was higher for the year 1 as compared to year2. Likewise, significantly higher grain yield was recorded in for genotype x in comparison to all other genotypes. As compared to controlled condition, heat stress reduced grain yield significantly. Moreover, GxE interaction was also significant and pronounced decrease in grain yield was recorded for genotype Y under heat stress conditions. Please follow same style for all other traits as well as significant interactions.
Also, include one small paragraph about broad-sense heritability
Discussion
Paragraphs of discussion are too long and too many in number. Only discuss the concrete results.
Round 2
Reviewer 2 Report
Dear Mohamed et al, many thanks for submitting the revised version of the manuscript. Why there are so many comments in the submitted version of the manuscript. Further, only few of the changes have been incorporated and most of comments are un-tackled. I will again suggest following changes in the manuscript.
Introduction
Please indicate a concrete hypothesis in the concluding paragraph to indicate how the sub-objectives were achieved.
Results
Overall, results should be written concisely. Current presentation does not make sense and can’t be eye catching. Please rewrite to express the meanings. For example, results of analysis of variance can be written as follow, grain yield was higher for the year 1 as compared to year2. Likewise, significantly higher grain yield was recorded in for genotype x in comparison to all other genotypes. As compared to controlled condition, heat stress reduced grain yield significantly. Moreover, GxE interaction was also significant and pronounced decrease in grain yield was recorded for genotype Y under heat stress conditions. Please follow same style for all other traits as well as significant interactions.
Also, include one small paragraph about broad-sense heritability
Discussion
Paragraphs of discussion are too long and too many in number. Only discuss the concrete results.
Round 3
Reviewer 2 Report
Still hypothesis in not given and I have no further comments
